# Tocilizumab Use among Patients Who Developed Pulmonary Embolism in the Course of Cytokine Release Storm and COVID-19 Pneumonia—A Retrospective Study

**DOI:** 10.3390/biomedicines10071581

**Published:** 2022-07-02

**Authors:** Daniel Chober, Bogusz Aksak-Wąs, Jolanta Niścigorska-Olsen, Małgorzata Niekrasz, Miłosz Parczewski

**Affiliations:** 1Department of Infectious, Tropical Diseases and Immune Deficiency, Pomeranian Medical University in Szczecin, 71-455 Szczecin, Poland; bogusz.aw@gmail.com (B.A.-W.); j.niscigorska-olsen@wp.pl (J.N.-O.); 2Department of Neurology, Regional Hospital in Szczecin, 71-455 Szczecin, Poland; mal.niekrasz@gmail.com

**Keywords:** COVID-19, tocilizumab, embolism, thrombosis, mortality

## Abstract

Introduction: Thromboembolic events, including mainly pulmonary embolisms and ischemic strokes, occur in up to one-third of COVID-19 patients. As efficacy of tocilizumab (TCZ) among patients with acute pulmonary embolism (PE) was not previously investigated, this study aimed to provide such data. Objectives: The aim of the study was to investigate the effect of TCZ on mortality in patients with confirmed acute pulmonary embolism, cytokine release storm and COVID-19 pneumonia. Patients and methods: Longitudinal data of 4287 patients with confirmed SARS-CoV-2 infection were collected between 4 March 2020 and 16 January 2022. In this study, we retrospectively analyzed the samples and dataset of cases with confirmed acute pulmonary embolism associated with at least moderate lung involvement due to COVID-19 pneumonia. Results: In the analyzed dataset, 64 adult patients were diagnosed with PE, and of these, 28 (44%) cases were treated with two 8 mg/kg doses of TCZ, and 36 (56%) did not receive this agent. The groups were balanced regarding demographics, comorbidities and the biochemical markers. Overall mortality in our study was 29.6% (n = 17). Mortality in the group treated with TCZ was 43% (n = 12) compared to 19% (n = 7) in the group without TCZ. In multivariate proportional Cox hazards models, intravenous administration of TCZ was independently associated with higher mortality (HR: 3.342 (CI: 1.077–10.370), *p* = 0.036). Conclusions: In patients with COVID-19 pneumonia with at least moderate lung involvement, CRS and acute pulmonary embolism, administration of TCZ is associated with increased mortality. Therefore, TCZ should be used with caution in SARS-CoV-2 cases with pulmonary embolism.

## 1. Introduction

Hemostasis and thrombosis are complex, multifactorial processes. Thrombosis, as a common mechanism underlying myocardial infarction, ischemic stroke and venous thromboembolism (VTE), is the leading global cause of mortality [1]. Structurally, arterial and venous thrombi notably differ: Arterial thrombi are rich in platelets and form in the circumference of ruptured plaques. Venous thrombi are rich in both fibrin and red blood cells and form even in the environment of the intact endothelial wall [2]. Pulmonary embolism (PE) affects 60–112 per 100,000 inhabitants/year [3,4]. According to prospective cohort studies, the acute phase mortality ranges from 7 to 11% [5,6,7].

Since the beginning of the severe acute respiratory syndrome coronavirus-2 (SARS-CoV-2) pandemic, nearly 430 million cases of this infection have been confirmed with over 6 million deaths worldwide [8]. While acute respiratory failure is the main reported cause of death in patients infected with SARS-CoV-2 [9], the studies conducted so far suggest the influence of an increased incidence of venous thrombosis, which significantly worsens the prognosis and increases mortality [10,11]. Meta-analyses indicate a two-fold increased risk of death in COVID-19 patients who developed a venous thrombotic event [12]. A plethora of studies and analyses have already confirmed a correlation between COVID-19 and the risk of thrombosis disclosing prothrombotic activity of this viral infection [13,14,15]. For example, a large meta-analysis reported that PE and deep vein thrombosis (DVT) were observed in 16.5% and 14.8% of COVID-19 patients, respectively, while in more than half of the patients with PE, no DVT was observed [16]. Furthermore, clinical data have shown an incidence of VTE in the COVID-19 patient population with a variable incidence between 7 and 31% [17,18].

Multiple risk factors contributing to the venous thrombosis in COVID-19 patients were identified, and include generalized hypercoagulability, dysfunctional immune response to SARS-CoV-2 leading to cytokine release syndrome (CRS), hyperinflammation associated with endothelial cell damage, hypoxemia resulting in cardiopulmonary failure, and metabolic disorders [19,20,21]. Immunothrombosis may be the main factor responsible for the development of thrombotic events in COVID-19 patients, as various types of innate immune activation have been observed concurrently in patients with such complications [22,23]. Hereditary diseases (such as the FVL mutation, fibrinogen gamma (FGG) mutation, homozygous prothrombin G20210A mutation) are also among the conditions favoring of thrombosis [24].

Anti-interleukin-6 receptor monoclonal antibody agent, TCZ, has been reported to reduce mortality among hospitalized patients with COVID-19 pneumonia [25,26]. TCZ should only be used in cases with identified CRS. CRS is a clinical diagnosis that is based on the presence of a fever (≥38.0 °C), with or without variable degrees of hypotension, hypoxia, and/or other end-organ dysfunctions. The temporal relationship to the triggering immune therapy is important for establishing the diagnosis of CRS. Laboratory studies are not required to diagnose CRS, but they may help to distinguish CRS from other conditions that can cause similar findings. CRS is associated with increased levels of IL-6. IL-6 levels exceeding 80 pg/mL are highly predictive of mechanical ventilation use [27]. There is limited evidence in the available literature on the effect of TCZ on the risk of thrombosis [28,29]. Isolated reports suggest an increased risk of thromboembolism (TE) associated with transient elevations in D-dimer levels [29], while other studies report a reduced risk of thrombosis in patients treated with TCZ [30]. Guidelines indicate that TCZ should only be used in cases with identified cytokine storm; however, the guidelines do not consider its safety among cases with thrombotic events [31]. Case reports suggest that physicians should be aware that tocilizumab may mask inflammatory markers (e.g., IL6, CRP, fibrinogen and ferritin) without reducing the risk of thrombotic events, creating a façade of better prognostic outcomes instead [28]. At the same time, other immunomodulatory drugs used in the cytokine storm phase of COVID-19, such as baricitinib, are clearly contraindicated among patients with increased thrombosis risk [32].

The aim of the study was to investigate the effect of TCZ on mortality in patients with confirmed acute PE and COVID-19 pneumonia. To our knowledge, this is a first study investigating safety of TCZ use among patients who developed PE in the course of COVID-19 before the introduction of immunomodulating treatment. Here, we analyze mortality risk and safety of TCZ in this selected population.

## 2. Materials and Methods

### 2.1. Study Groups

Our database contains information about patients hospitalized at the Regional Hospital in Szczecin, Poland. Patients participating in the study were observed from 4 March 2020 to 23 January 2022 when the database was closed. The database contains 4287 cases. Informed consent for data analysis was obtained from all subjects participating in the study. All data were fully anonymized before statistical analyzes.

In this study, we retrospectively analyzed the dataset of patients with confirmed acute PE associated with at least moderate lung involvement due to COVID-19 pneumonia. Patients enrolled in this study before admission to hospital reported drops in oxygen saturation less than or equal to 90% and fever (>38 °C). In each case, SARS-CoV-2 infection was confirmed using SARS-CoV-2 polymerase chain reaction (PCR) tests from throat swabs. In each case, pneumonia was confirmed using computed tomography of the chest (C-CT). All laboratory and CT (including angio-CT of pulmonary arteries) tests were performed immediately upon admission, within the first 24 h.

The primary objective of this study was to determine the efficacy and safety of TCZ in patients with COVID-19 pneumonia and acute PE. A secondary goal of this study was to try to identify the risk factors associated with higher mortality. In order to answer the above questions, we established the following inclusion and exclusion criteria for the study:Inclusion criteria for this study were as follows:
-CT confirmed COVID-19-pneumonia with at least 10% lung involvement;-CT and clinically proven acute PE;-CRS with IL-6 levels exceeding 80 pg/mL;-Age > 18 years;-Fever > 38 °C at admission to hospital;-Oxygen saturation less or equal to 90%;-Confirmed ongoing SARS-CoV-2 infection.Exclusion criteria:
-Procalcitonin level > 2 ng/mL-as a predictor of septic complications where TCZ is contraindicated;

We divided all analyzed patients into two groups. The first group consisted of patients who received 8 mg/kg intravenous tocilizumab (maximum dose: 800 mg) twice, 12 h apart. The second group consisted of patients who did not consent to the administration of tocilizumab and patients who, despite consenting, did not receive drugs due to medical contraindications or for non-medical reasons (no drugs available in the hospital). The second group used only standard treatment in line with the guidelines. Before administering tocilizumab, we ruled out acute viral infections including cytomegalovirus, hepatitis B and C as well as human immunodeficiency virus based on standard serological methods. Acute toxoplasma gondii was also excluded prior to TCZ administration. The median time from hospital admission to administration of first dose of tocilizumab was 1 day (IQR 1–2 days). According to the guidelines of the Polish Society of Epidemiologists and Doctors of Infectious Diseases and the Summary of Product Characteristics, a second dose of TCZ may be administered if the general condition deteriorates or there is no improvement. Due to the lack of clearly defined criteria for improving the patient’s general condition, we assumed that all patients who persistently had dyspnea or fever should be given another dose of TCZ.

The endpoint for this analysis was death or discharge from the hospital.

To evaluate lung involvement, we used the scale recommended by the French Chest Imaging Society (SIT), the European Society of Radiology (ESR) and the European Society for Chest Imaging (ESTI). Following the recommendations of SIT and ESTI, we used the following division of pneumonia: (1) absent or minimal (<10%), (2) moderate (10–25%), (3) extensive (25–50%), (4) severe (50–75%), and (5) critical (>75%) [33,34].

Decision on the hospital admission was based on the assessment of the doctors on duty working in the Hospital Emergency Department and the Infectious Diseases Admission Room. Treatment was in line with the current knowledge, guidelines of the Polish Society of Epidemiologists and Infectious Diseases Specialists (PTEiLChZ) and product characteristics [31,35].

If given, the following concomitant medications were used:

The decision to use remdesivir has always been made by a physician. Remdesivir was administered in 5-day courses. On day 1, it was given as a 200 mg loading dose followed by a 100 mg maintenance dose thereafter. The first administration of remdesivir was started within 7 days of onset of symptoms, if symptoms lasted longer, treatment was not started.

-Chloroquine was not used.-Supportive treatment was applied to each of the patients.The supportive treatment included:-antibiotic therapy (Ceftriaxon was drug of choice but could vary depending on the patient’s condition);-Oxygen therapy (Low-/high-flow oxygen therapy or mechanical ventilation were used. No ECMO were used);-Intravenous rehydration;-Dexamethasone administered intravenously in a dose of at least 6 mg per day;-Low-molecular-weight or non-fractionated heparin in therapeutic doses.All patients, in accordance with the applicable standards and guidelines, received:-Prophylactic antibiotic (preferably ceftriaxone in a dose of at least 2 g per day);-Glucocorticoids (dexamethasone in a dose of at least 6 mg per day iv);-Supportive oxygen supplementary therapy;-Intravenous rehydration.

In most cases, the existing concomitant drugs were maintained, unless they required modification, e.g., in diabetic patients, oral hypoglycemic drugs were abandoned in favor of insulin according to the glycemic profile. None of the patients required thrombolytic therapy including administration of alteplase, streptokinase, and urokinase.

### 2.2. Ethical Issues

The study protocol was approved by the Bioethical Committee of Pomeranian Medical University, Szczecin, Poland (approval number: KB-0012/92/2020). All patients provided informed consent for participation in the study, related with administration of off-label drug—tocilizumab. The study was conducted in accordance with principles of the Declaration of Helsinki.

### 2.3. Sampling and Data Collection Methodology

Clinical data such as gender, age, comorbidities, treatment history, length of stay in hospital, duration of ICU treatment, survival statistics, chest computed tomography results, baseline blood oxygenation levels, and selected laboratory parameters were collected from medical records. Among the selected laboratory tests were lymphocytes and red blood cells count, hemoglobin levels, platelet count, procalcitonin levels, C-reactive protein levels, interleukin 6 levels, lactate dehydrogenase levels, d-dimer activity, eGFR and aspartate and alanine aminotransferase activity. All cases of PE were confirmed radiographically using CT angiography. Comorbidities were assessed in terms of their incidence.

### 2.4. Statistics

Baseline laboratory parameters and clinical data were calculated separately for parametric and non-parametric statistics. We used the Mann–Whitney U test for nonparametric statistics to perform a statistical comparison. The multiway tables were constructed taking into account the chi-square test. Kaplan–Meyer cumulative mortality was calculated with statistical significance of survival data analyzed using log-rank test. Unadjusted and multivariate Cox proportional hazards models were used to assess the impact of the analyzed parameter on the risk of death and to calculate the risk factors (HR). The Akaike’s information criteria was base for selection of the final model. The *p*-values of 0.05 were considered significant. For the statistical calculations, we used commercial software (Statistica 13.0 PL; Statasoft, Warsaw, Poland).

## 3. Results

### 3.1. Clinical Characteristics of Patients with COVID-19

From our database, we selected cases accordingly to inclusion and exclusion criteria as showed on flowchart (Figure 1).

The final dataset included 64 adult patients, median age 68 (IQR 57–75) years, with confirmed acute PE (Table 1). All cases were classified as moderate, extensive, severe or critical lung involvement based on chest computed tomography (C-CT) and the median of lung involvement was 38.27% (IQR 24.77–50.34). The key biochemical parameters indicative of advanced inflammation were significantly elevated. The median IL-6 was 138.50 pg/mL (IQR 113.00–206.50) with a concurrently low procalcitonin concentration (median 0.17 ng/mL (IQR 0.10–0.31)). Lactate dehydrogenase activity was high in most of the cases (median 499.50 U/L (IQR 349.00–638.00)), which is known to be associated with severe course of the disease. The most common concomitant disease was arterial hypertension, which was present in 13 (20%) cases. Diabetes was found in two cases (3%), and a history of cancer in one case (1%).

### 3.2. Clinical and Laboratory Data on TCZ Treated vs. Control Group

The study group included 28 (44%) patients who received TCZ and 36 (56%) controls treated with standard of care only. There were no statistically significant differences between particular groups regarding age, percentage of lung involvement, gender, comorbidities, white blood cells count, red blood cells count, hemoglobin level, hematocrit, platelets count, procalcitonin levels, C-reactive protein levels, interleukin-6 levels, lactate dehydrogenase levels, D-dimer activity, creatinine levels, eGFR, aspartate and alanine aminotransferase activity, bilirubin levels, troponin T and CKMB levels (Table 2).

Median length of hospital stay was 17 (IQR 12.5–24.0) days and did not differ between the TCZ receiving group (17.0 (IQR 12.5–25.5) days) compared to (17.0 (IQR 12.5–22.0) days) for non TCZ [*p* = 0.645].

Only six (9%) patients were treated in ICU. All patients in ICU were on invasive mechanical ventilation. The low percentage of ICU admissions was associated with the constant lack of beds in the ICU, age-related disqualification and the phenomenon of “sudden death”, when the deterioration of vital functions occurs so quickly that it is impossible to transfer to the ICU.

### 3.3. Overall Mortality Risk

Out of 64 patients, 19 (30%) died from COVID-19-related pneumonia and its complications and 45 (70%) survived. Median length of hospitalization for the group who did not survive was 12 (IQR 4–20) days. Median length of hospitalization for the group who survived was 20 (IQR 14–26) days. (See Figure 2.)

We also analyzed the group in the context of death risk factors, regardless of the TCZ use (Table 3). Higher mortality factors in our analysis were associated with age and procalcitonin T levels. Other factors did not differ significantly (Table 3).

### 3.4. Tocilizumab-Associated Mortality

Mortality in the TCZ group was 43% (n = 12) compared to 19% (n = 7) in the non-TCZ group.

We constructed multivariate Cox proportional hazards models for all statistically significant factors and for the administration of TCZ in line with the main goal of our study.

We included the use of Remdesivir in our calculations to estimate the potential impact of other drugs on survival in patients with PE. In multivariate proportional Cox hazards models (Table 4), age (HR: 1.118 (CI: 1.055–1.187), *p* = 0.001), and administration of TCZ was associated with higher mortality (HR: 3.342 (CI: 1.077–10.370), *p* = 0.036). (See Table 4.)

## 4. Discussion

To our knowledge, this is the first study investigating the relationship between Tocilizumab administration and the risk of mortality among patients who developed PE in the course of CRS and COVID-19 pneumonia. Our study shows that the use of tocilizumab in patients with acute PE correlates with an increased risk of death.

It is well known that all patients hospitalized for severe acute infectious disease are at a higher risk of thromboembolic events [18]. Evidence strongly highlighted the fact that severe COVID-19 pneumonia is often complicated by coagulopathy, which significantly increased the risk of acute PE and thus mortality [12]. According to some studies, TE is associated with COVID-19 pneumonia with a frequency of 20–30% [36,37], reaching up to 40–70% [38,39,40]. At the same time, correlation between the occurrence of TE and the severe course of the disease, including admission to the intensive care unit and potential death, has been regularly reported [41]. Mortality in the group of patients with both proximal and distal DVT was significantly higher compared to the group without DVT [41]. However, no differences in mortality were found between proximal DVT and distal DVT [41]. Numerous studies have shown that in-hospital treatment with heparin was associated with a lower mortality, especially in critically ill COVID-19 patients and in those with highly activated blood clotting [42,43].

For this study, we have selected a unique, high-risk group of COVID-19 patients with diagnosis of acute PE in whom the biochemical parameters of CRS were established. With no statistically significant differences between particular groups regarding age, comorbidities, and initial biochemical markers, we found significantly higher 30-day mortality [HR: 3.342 (CI: 1.077–10.370), *p* = 0.036] among patients treated with TCZ in addition to standard treatment as compared to those treated with standard treatment (dexamethasone, antibiotics, intravenous rehydration, supportive oxygen supplementary therapy). The difference suggests a negative influence of TCZ on the applied anticoagulation therapy.

In Surbhi Warrior’s study, a lower incidence of thrombosis in COVID-19 patients who received steroids together with tocilizumab was reported. The database for this study consisted of 1265 patients, of whom only 4.8% (N = 61) had been diagnosed with PE. This study suggests that steroids and tocilizumab may reduce the proinflammatory state that leads to thrombosis [30]. At the same time, it was emphasized that mortality was higher in patients who received COVID-19-related treatment. Administration of TCZ was associated with an increase in the odds ratio of mortality (OR 2.51 (CI 1.06–5.96)). The authors of the study assumed that this was related to the severity of the disease. In comparison, our data indicate that the severity of the disease in both groups is similar and that the negative effect of TCZ on survival may be independent. Our study database was significantly larger.

Kok Hoe Chan’s study showed a relationship between the concentration of IL-6 and venous thrombotic disease. The database for this study consisted of 436 patients, of whom only 24 met the inclusion criteria. Transient elevations in D-dimer levels have been reported in patients with COVID-19 who received TCZ. The trend towards an increased number of deaths related to TE was highlighted [29]. Our studies differ in terms of eligibility for the study. The only admission criteria for the above-mentioned study were confirmed SARS-CoV-2 virus infection and treatment with tocilizumab. For comparison, in our study we narrowed the group to people with radiologically confirmed acute PE. Moreover, our group is much more numerous and homogeneous in terms of laboratory tests and the CT image.

Conversely to our data, Gagan Kumar’s study concluded that tocilizumab was not associated with VTE [44]. The database for the cited study consisted of 4645 patients, of whom only 91 had been diagnosed with PE. Overall, 251 (5.4%) patients had VTE. However, in this study, the only inclusion criterion was confirmed SARS-CoV-2 virus infection in people over 18 years of age. Moreover, only 14.3% (n = 36) of patients with VTE were treated with tocilizumab. There were no data on the incidence and correlation of PE in the above-mentioned study.

Our study has several limitations, mostly related to its observational and retrospective nature of the presented analysis, which should be confirmed by prospective data. Immunomodulating treatment was selected by physicians on the basis of knowledge and guidelines, periodically limited by the availability of drugs and without randomization.

The main advantage of the current analysis is the large initial dataset, which allows for homogeneous analysis of at-risk patient groups. Collection of data from one center allowed us to exclude the possibility of an error related to different standards of individual tests. Another advantage of our analysis is the accurate assessment of lung involvement and its impact on the treatment process. Last but not least, rigorous inclusion criteria for the study allowed for the exclusion of an error resulting from a poorly selected study group.

## 5. Conclusions

For patients with acute PE, with radiologically proven at least moderate COVID-19-related lung involvement, the administration of a double dose of TCZ together with standard treatment (based on dexamethasone, antibiotic therapy) provides a much worse effect than the standard treatment alone in terms of survival. A dysfunctional immune response to SARS-CoV-2 leading to a CRS may contribute to the earlier onset of acute ischemic stroke or PE. Tocilizumab is the drug of choice for patients requiring immunomodulatory therapy to reduce excessive cytokine production. However, it should be used with extra caution, especially in patients with a confirmed episode of TE. Only further research into understanding the essence of CRS and possible immunomodulatory therapies can help to solve this problem.

## Figures and Tables

**Figure 1 biomedicines-10-01581-f001:**
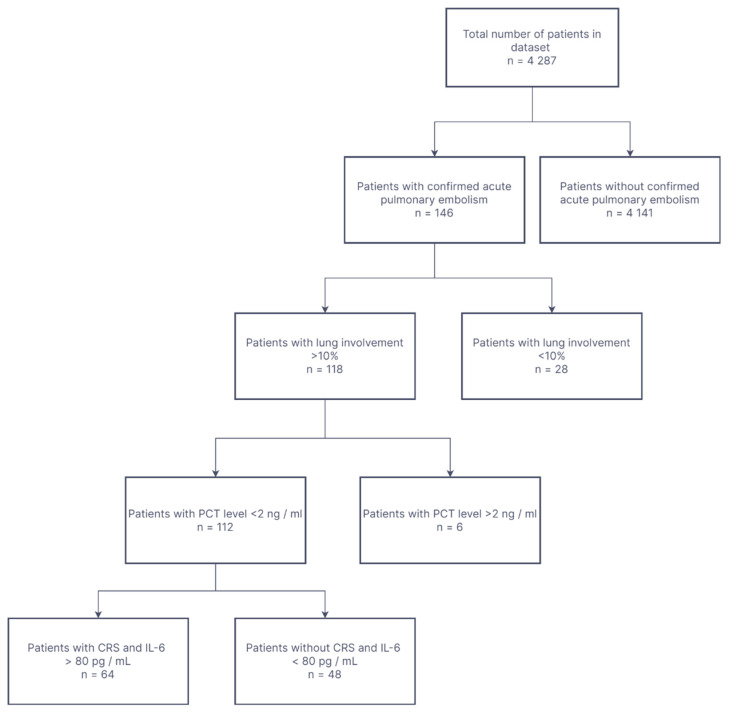
Study flow diagram.

**Figure 2 biomedicines-10-01581-f002:**
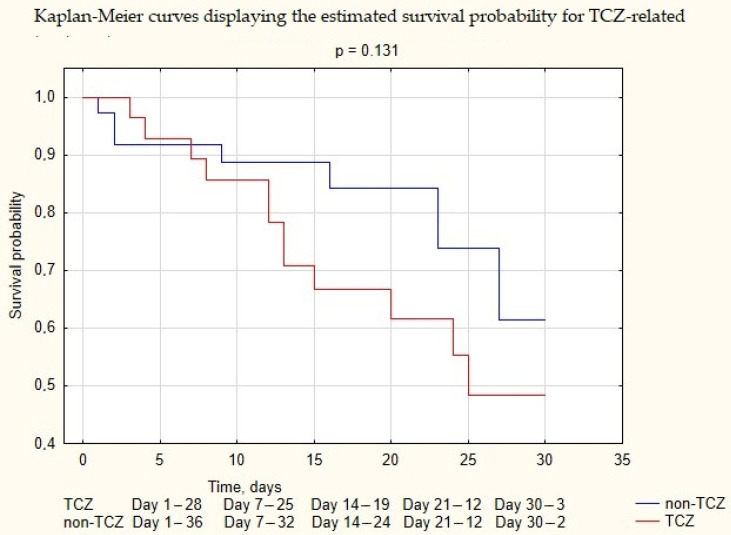
Kaplan-Meier curves displaying the estimated survival probability for TCZ-related treatment.

**Table 1 biomedicines-10-01581-t001:** Baseline characteristics of all patients included in the study.

	Baseline Characteristics of All Patients Included in the Study			
	N	Median	Lower Quartile	Upper Quartile
Age, years	64	68.00	57.00	75.00
Percentage of Lung Involvement,%	64	38.27	24.77	50.34
WBC, ×10^3^/μL	64	8.81	6.72	10.91
RBC, ×10^6^/μL	64	4.66	4.25	5.06
HGB, g/dL	64	13.95	12.65	15.10
HCT,%	64	40.15	36.85	43.45
Platelets, ×10^3^/μL	64	227.50	179.00	302.50
Procalcitonin, ng/mL	63	0.17	0.10	0.31
CRP, mg/L	64	146.40	104.68	205.53
IL-6, pg/mL	64	138.50	113.00	206.50
LDH, U/L	58	499.50	349.00	638.00
D-dimer, ug/L	64	1611.50	631.50	8481.50
Creatinine, mg/dL	64	0.94	0.82	1.22
eGFR, mL/min	64	80.25	57.61	96.21
AST, U/L	57	49.00	35.00	72.00
ALT, U/L	57	36.00	22.00	62.00
Glucose, mg/dL	61	123.00	113.00	145.00
Bilirubin total, mg/dL	59	0.63	0.41	0.85

**Table 2 biomedicines-10-01581-t002:** Characteristic of Groups TCZ vs. Non-TCZ.

	TCZ N = 28 (44%)	Non-TCZ N = 36 (56%)	*p* Value
Age, years	68.5 (54.5–75)	67 (61.5–74)	0.538
Percentage of Lung Involvement,%	33.82 (24.28–49.22)	43.67 (28.2–53.34)	0.211
WBC, ×10^3^/μL	8.43 (6.65–10.78)	8.81 (7.39–10.96)	0.461
RBC, ×10^6^/μL	4.58 (4.13–5.01)	4.76 (4.45–5.13)	0.206
HGB, g/dL	13.6 (11.75–15.2)	14.05 (13.05–15.05)	0.218
HCT,%	39.95 (35–43.45)	41.3 (38.85–43.65)	0.307
Platelets, ×10^3^/μL	227.5 (176–316.5)	232 (180–290)	0.898
Procalcitonin, ng/mL	0.17 (0.1–0.35)	0.2 (0.12–0.3)	0.429
CRP, mg/L	132.54 (97.64–227.02)	153.93 (132.29–189.51)	0.201
IL-6, pg/mL	137.5 (104.55–209)	146.5 (116–206.5)	0.844
LDH, U/L	481 (355–638)	515 (345–674)	0.919
D-dimer, ug/L	2301.05 (695–8481.5)	912.5 (586–6799)	0.486
Creatinine, mg/dL	0.95 (0.74–1.23)	0.94 (0.88–1.22)	0.477
eGFR, mL/min	80.39 (56.66–103.63)	79.47 (58.98–90.25)	0.574
AST, U/L	53 (41–82)	47.5 (34–69)	0.378
ALT, U/L	43 (22–89)	33 (25–55)	0.665
Glucose, mg/dL	123 (114–145)	123.5 (112–141.5)	0.739
Bilirubin total, mg/dL	0.7 (0.45–0.98)	67 (61.5–74)	0.205
Gender, n (%) Male (reference)	24 (86%)	23 (64%)	Chi-square Pearson *p* = 0.049
Diabetes, n (%) Yes (reference)	2 (7%)	0 (0%)	Chi-square Pearson *p* = 0.103
Hypertension, n (%) Yes (reference)	5 (18%)	8 (22%)	Chi-square Pearson *p* = 0.666
Cancer, n (%) Yes (reference)	0 (0%)	1 (3%)	Chi-square Pearson *p* = 0.374
Remdesivir admission, n (%) Yes (reference)	23 (82%)	28 (78%)	Chi-square Pearson *p* = 0.666
ICU admission, n (%) Yes (reference)	4 (14%)	2 (6%)	Chi-square Pearson *p* = 0.234
Mortality, n (%) Died (reference)	12 (43%)	7 (19%)	Chi-square Pearson *p* = 0.041

**Table 3 biomedicines-10-01581-t003:** Characteristic of surviving group compared to the patients who died.

	Survived N = 45(70%)	Died N = 19 (30%)	*p* Value
Age, years	67 (54–71)	76 (63–82)	0.002
Percentage of Lung Involvement,%	35.67 (23.88–49.49)	44.77 (30.49–60.57)	0.113
WBC, ×10^3^/μL	8.59 (6.67–10.93)	9.34 (7.09–10.77)	0.730
RBC, ×10^6^/μL	4.78 (4.32–5.15)	4.46 (4.14–4.81)	0.092
HGB, g/dL	14.1 (12.9–15.2)	13.3 (12.1–14.4)	0.123
HCT,%	40.9 (37.3–43.6)	39.1 (34.4–42.4)	0.169
Platelets, ×10^3^/μL	242 (183–315)	224 (178–288)	0.370
Procalcitonin, ng/mL	0.16 (0.1–0.25)	0.27 (0.14–0.49)	0.028
CRP, mg/L	138.6 (101.63–188.16)	163.83 (117.35–207.23)	0.258
IL-6, pg/mL	129 (95.7–176)	168 (123–242)	0.071
LDH, U/L	503.5 (355–636)	474 (313–678)	0.608
D-dimer, ug/L	1223 (603–8463)	2145 (805–8861)	0.436
Creatinine, mg/dL	0.94 (0.8–1.1)	1.03 (0.83–1.55)	0.246
eGFR, mL/min	83.11 (66.38–96.55)	72.94 (41.61–93.58)	0.146
AST, U/L	53 (41–77)	38 (30–69)	0.160
ALT, U/L	43.5 (28–79)	26 (16–55)	0.084
Glucose, mg/dL	121.5 (111–136)	127 (114–173)	0.327
Bilirubin total, mg/dL	0.64 (0.41–0.83)	0.62 (0.36–0.96)	0.651
Gender, n (%) Male (reference)	34 (75%)	13 (68%)	Chi-square Pearson *p* = 0.554
Diabetes, n (%) Yes (reference)	2 (4%)	0 (0%)	Chi-square Pearson *p* = 0.350
Hypertension, n (%) Yes (reference)	12 (27%)	1 (5%)	Chi-square Pearson *p* = 0.052
Cancer, n (%) Yes (reference)	1 (2%)	0 (0%)	Chi-square Pearson *p* = 0.512
Remdesivir admission, n (%) Yes (reference)	34 (76%)	17 (90%)	Chi-square Pearson *p* = 0.206
ICU admission, n (%) Yes (reference)	2 (4%)	4 (21%)	Chi-square Pearson *p* = 0.037

**Table 4 biomedicines-10-01581-t004:** Cox proportional-hazards model for mortality risk.

	Cox Proportional-Hazards Model for Mortality			
	*p* Value	Hazard Ratio (HR)	Lower 95%CI HR Value	Upper 95%CI HR Value
Age, years	0.001	1.119	1.055	1.187
Percentage of Lung Involvement,%	0.511	1.009	0.980	1.039
Procalcitonin, ng/mL	0.084	2.039	0.906	4.586
Administration of Remdesivir (reference) vs. no Remdesivir	0.132	0.261	0.045	1.502
Administration of TCZ (reference) vs. no TCZ	0.036	3.342	1.077	10.370

## Data Availability

The original anonymous dataset is available on request from the corresponding author at Daniel.chober@pum.edu.pl.

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
