# Peer review of "Tocilizumab Use among Patients Who Developed Pulmonary Embolism in the Course of Cytokine Release Storm and COVID-19 Pneumonia—A Retrospective Study"

_biomedicines, 2022, doi:10.3390/biomedicines10071581_

Round 1
Reviewer 1 Report
This study tried to elucidate the impact of tocilizumab (TCZ) in COVID-19 patients with pulmonary embolism. However, several questions must be clarified in advance.
1. In the protocol, TCZ was given at a dose of 8mg/kg twice, 12 hours apart (page 3, line 24). All currently available guidelines suggested only a single dose of TCZ. Is there any reference for the routine use of double doses in COVID-19 patients? If so, will it be double dose TCZ itself harmful to the patients? Can we still conclude that TCZ is harmful to patients with pulmonary embolism, or maybe, “double dose TCZ is harmful in patients with pulmonary embolism”?
2. Page 2, line 23: Cytokine storm syndrome (CRS) is diagnosed clinically. We agreed that an increase in IL-6 is associated with CRS. However, is there any reference that shows that we can simply use IL-6 > 80pg/mL to diagnose CRS (Figure 1)? The cited reference only mentioned that patients with IL-6 more than 80pg/mL is highly predictive of ventilator use, but not CRS.
3. Typos were found in Table 1, Table 2 and Table 3, such as “patelets” and “admision”. Besides, the last row of Table 2 is confusing; the number denoted “mortality rate” or “survival rate”?
4. Abbreviations should all be checked, and the full form should be stated at the first time. An example is SARS-CoV-2, first appears in page 1 line 43, but full form was stated in page 3 line 1. CRS was mistyped as CRP in page 2 line 24. What is TE in page 9, line 4? And “COIVD-19” was noticed in the “conclusion” paragraph.
5. In paragraph 3.3 (page 7, line 6), please explain what is meant by “length of treatment” and what kind of treatment is it.
6. In Figure 2, the final survival probability of the non-TCZ group was between 0.6-0.7, and the p value of the analysis was 0.131. Is there any reason why the number is differed from the result of other statistics?
7. Several confounding factors should be aware. The study was conducted from 2020 to 2022, and as we know, the virulence and prevalent strains of SARS-CoV-2 varied throughout the period. Will the higher mortality be associated with the strain of SARS-CoV-2 rather than the treatment? Is there any relationship between mortality with previous COVID-19 vaccination of the patient?
8. Many factors could affect mortality in patients with pulmonary embolism. The author mentioned very few regarding the severity, clinical course, and treatment for pulmonary embolism. Besides, although all patients can be classified as at least severe COVID-19 (SpO2 < 94% in room air; 90% was applied as inclusion criteria in this study) according to WHO classification, the study did not mention any complications such as mechanical ventilation and shock, could these factors affect the high mortality in the TCZ group? Can we solely use laboratory data to represent the severity of the patient?
Author Response
Dear Reviewer,
we would like to very warmly thank the reviewer for the diligent and the most valuable comments. We have taken every effort to include them accurately in the manuscript, and hope that the manuscript is suitable for further processing and publication in Biomedicines.
Below please find responses to the specific comments, also highlighted in the text.
With kindest regards,
Daniel Chober
- According to summary of product characteristics:
“The recommended posology for treatment of COVID-19 is a single 60-minute intravenous infusion of 8 mg/kg in patients who are receiving systemic corticosteroids and require supplemental oxygen or mechanical ventilation. If clinical signs or symptoms worsen or do not improve after the first dose, one additional infusion of RoActemra 8 mg/kg may be administered. The interval between the two infusions should be at least 8 hours.”According to the guidelines of the Polish Society of Epidemiologists and Doctors of Infectious Diseases:
“Tocilizumab (in patients with IL-6 >100 pg/ml) is used intravenously in a single dose of 800 mg if PBW >90 kg; 600 mg if PBW >65 kg and ≤90 kg; 400 mg if PBW >40 kg and ≤65 kg; and 8 mg/kg if PBW ≤40 kg. The second dose can be administered 8–24 h after the first one, if the patient’s general condition has not improved”
We have added the following information to the 2.1 section:
According to the guidelines of the Polish Society of Epidemiologists and Doctors of Infectious Diseases and the Summary of Product Characteristics, a second dose of tocilizumab may be administered if the general condition deteriorates or there is no improvement. Due to the lack of clear defined criteria for improving the patient's general condition, we assumed that all patients who persistently dyspnea or fever should be given another dose of tocilizumab.
We have changed the sentence in section 5. Correct reference has been added:
For patients with acute pulmonary embolism with radiologically proven at least moderate COIVD-19-related lung involvement administration of double dose of tocilizumab together with standard treatment (based on dexamethasone, antibiotic therapy) provides much worse effect then the standard treatment alone in terms of survival
- We have changed the sentence. Correct reference has been added:
CRS is a clinical diagnosis that is based on the presence of a fever (≥38.0°C), with or without variable degrees of hypotension, hypoxia, and/or other end-organ dysfunctions. The temporal relationship to the triggering immune therapy is important for establishing the diagnosis of CRS. Laboratory studies are not required to diagnose CRS, but they may help to distinguish CRS from other conditions that can cause similar findings. CRS is associated with increased levels of IL-6. IL-6 levels exceeding 80 pg / ml are highly predictive of mechanical ventilation use.
Flowchart (Figure 1.) has been changed as well.
- All typos were corrected. Last row in Table 2. was corrected.
- Abbreviations were checked and corrected.
- We have changed the sentence. Correct reference has been added:
Median length of hospitalization for the group who did not survive was 12(IQR 4-20) days. Median length of hospitalization for the group who survived was 20 (IQR 14-26) days. (Figure 2.)
- The Kaplan–Meier estimator also known as the product limit estimator, is a non-parametric statistic used to estimate the survival function from lifetime data while other statistic are used in tables 1-3 were parametric Chi Square and they cant be compared.
- We do not have information on vaccinations taken among patients, so we cannot rule out their impact on mortality. The situation is similar with regard to strains of SARS-CoV-2. However, the homogeneity of the patients in terms of the percentage of lung involvement, the results of the laboratory tests, indicates a similar degree of disease severity.
- Our study mention mechanical ventilation – only 6 patients were admitted to the ICU and required mechanical ventilation. 4 from the TCZ group and 2 from the non-TCZ group. The difference is not statistically significant and therefore we can conclude that it did not affect the higher mortality in the TCZ group.
In addition to the laboratory results, we used a radiographically determined percentage of lung involvement in the assessment of the severity of the patients' condition. The combination of laboratory and imaging tests seems to be the most objective way to assess the severity of the disease.
Reviewer 2 Report
This study investigates about the possible role of tocilizumab in influencing mortality in thrombotic events during CID 19 infection.
Although with some limitation, nevertheless recognized by the authors themselves, the value of the paper is clear. The reported negative effects of tocilizumab bring some new information that will surely prompt further studies that will definitively clarify the issue We just suggest to the authors to add, among the conditions favouring thrombosis also inherited conditions: Severe systemic thrombosis in a young COVID-19 patient with a rare homozygous prothrombin G20210A mutation. Infez Med. 2021 Jun 1;29(2):259-262.
Author Response
Dear Reviewer,
we would like to very warmly thank the reviewer for the diligent and the most valuable comments. We have taken every effort to include them accurately in the manuscript, and hope that the manuscript is suitable for further processing and publication in Biomedicines.
Below please find responses to the specific comments, also highlighted in the text.
With kindest regards,
Daniel Chober
- We have added sentence:
Hereditary diseases (such as the FVL mutation, fibrinogen gamma (FGG) mutation, homozygous prothrombin G20210A mutation) are also among the conditions favoring of thrombosis. [24]
Reviewer 3 Report
Strenght:
in this study Longitudinal data of 4,287 patients suffering covid 19 were studied regarding the efficacy of tocilizumab (TCZ) among patients with acute pulmonary embolism (PE), and mortality in patients with confirmed acute pulmonary embolism, cytokine release storm and COVID-19 pneumonia. In patients with COVID-19-pneumonia with at least moderate lung involvement, CRS and acute pulmonary embolism, administration of TCZ is associated with increased mortality, so TCZ should be used with caution in SARS CoV-2 cases with pulmonary embolism. In this paper several valuable graphics and tables are provided for phisicians.
Weakness: please check accurately english language
Author Response
Dear Reviewer,
we would like to very warmly thank the reviewer for the diligent and the most valuable comments. We have taken every effort to include them accurately in the manuscript, and hope that the manuscript is suitable for further processing and publication in Biomedicines.
Below please find responses to the specific comments, also highlighted in the text.
With kindest regards,
Daniel Chober
1. We have checked and corrected typos and grammar errors.
Round 2
Reviewer 1 Report
Thank you for the reply, and only a few questions left.
1. It is better to mention "two 8mg/kg doses of tocilizumab was given" in the abstract, not only in the conclusion section.
2. A concise and clear text can help the readers to understand the study. Please recheck and unify the abbreviations in the text, e.g., "SARS-COV-2" and "SARS-CoV-2" both presented in the manuscript (we suggest using the latter), "tocilizumab (TCZ)" appears twice in the abstracts, and there are many forms of tocilizumab, including "Tocilizumab", "tocilizumab", "TCZ" in the text.
3. As stated in the reply 8, the author only mentioned ICU admission in the manuscript, but it does not equal mechanical ventilation. A more explicit statement is needed for the readers to understand the severity of the patient group. On the other hand, since the mortality cases outnumbered the cases receiving ventilators and ICU stay, is there any explanation that patients with such severity who finally passed away did not receive intensive care, including intubation? Since mortality is the primary outcome in this study, any confounding factors that can affect mortality should be accounted for before we can conclude that two doses of tocilizumab can be harmful.
